

# Effects of prophylactic nebulized antibiotics on the prevention of ICU-acquired pneumonia: a systematic review and meta-analysis

Ming Gao, Xiaoxu Yu, Xiaoxuan Liu, Yuan Xu, Hua Zhou and Yan Zhu

Department of Critical Care Medicine, Beijing Tsinghua Changgung Hospital, School of Clinical Medicine, Tsinghua University, Beijing, China

Corresponding author
Yan Zhu, zya02226@btch.edu.cn

## ABSTRACT

**Objective:** To evaluate the efficacy and safety of prophylactic nebulized antibiotics in preventing intensive care unit (ICU)-acquired pneumonia through a meta-analysis.
**Methods:** Randomized controlled trials (RCTs) investigating the potential reduction in the incidence of ICU-acquired pneumonia through prophylactic nebulized antibiotics were collected by searching the PubMed, Embase, and Cochrane Library databases from their inception to January 23, 2024. The primary endpoint was the incidence of ICU-acquired pneumonia, while the secondary endpoints included mortality, length of ICU stay, mechanical ventilation days, and nebulization-related side effects. Statistical analyses were performed using RevMan 5.3 and STATA 14.0 software.
**Results:** A total of six RCTs were included in the analysis, involving 1,287 patients (636 patients in the study group received prophylactic antibiotic therapy, including Polymyxin B, Tobramycin, Ceftazidime, Colistimethate sodium, and amikacin; 651 patients in the control group primarily received saline). The results indicated that prophylactic nebulized antibiotic therapy significantly reduced the incidence of ICU-acquired pneumonia compared to that in the control group (odds ratio (OR) = 0.57, 95% confidence interval (CI) [0.43–0.74], $P < 0.0001$). No significant difference was observed in the mortality rate between the treatment and control groups (OR = 0.86, 95% CI [0.68–1.10], $P = 0.24$). Prophylactic nebulized antibiotic therapy also did not significantly reduce the length of ICU stay (MD = 0.2 days; 95% CI [−0.81 to 1.20], $P = 0.70$) or the number of mechanical ventilation days (MD = 0.43 days; 95% CI [−0.47 to 1.33], $P = 0.35$). Additionally, there was no evidence that prophylactic nebulized antibiotic therapy contributed to the development of multiple drug-resistant (MDR) bacterial pneumonia or increased the incidence of associated side effects, such as airway spasms.
**Conclusions:** This meta-analysis suggests that ICU-acquired pneumonia can be prevented by prophylactic nebulized antibiotic therapy in critically ill patients without increasing the risk of MDR bacterial infections or airway spasms. However, the reduction in the incidence of ICU-acquired pneumonia did not result in significant improvements in mortality or length of ICU stay.

## INTRODUCTION

Intensive care unit (ICU)-acquired pneumonia is a common complication in critically ill patients admitted to the ICU, regardless of their intubation status. This condition is associated with factors such as biofilm formation during tracheal intubation of critically ill patients, inhibition of the cough reflex, and a reduction in mucociliary clearance ability (*Dodek et al., 2004*; *Kollef, 1999*). The decrease in local defense mechanisms may be caused by the invasion of opportunistic pathogens into the lower respiratory tract or may be related to long-term bed rest, chest trauma, or other reasons. The occurrence of ICU-acquired pneumonia leads to an increase in the length of ICU stay and patient mortality, significantly increasing the economic burden on patients (*Warren et al., 2003*; *Ehrmann et al., 2023*). Numerous scholars have attempted to mitigate its occurrence through various interventions, such as reducing upper respiratory tract colonization, employing selective oral or intestinal decontamination, minimizing the use of proton pump inhibitors (PPIs), and utilizing endotracheal tubes with subglottic suction for intubated patients (*Lorente et al., 2007*; *Lacherade et al., 2010*; *Shibli, Milbrandt & Baldisseri, 2010*). However, the impact of these measures on reducing the incidence of ICU-acquired pneumonia has been found to be minimal.

In recent years, nebulized antibiotics have been increasingly valued by doctors for the treatment of severe pneumonia (*Sweeney & Kalil, 2019*). Compared with the intravenous application of antibiotics, nebulized antibiotics offer rapid and concentrated delivery directly to the treatment site, resulting in a high pulmonary-to-plasma concentration ratio. This targeted approach can potentially minimize the systemic side effects commonly associated with antibiotic use, such as liver and kidney damage and *Clostridioides difficile* infection. Inhaled antibiotic therapy has been widely used in the clinical treatment of ventilator-associated pneumonia. However, it remains unclear whether prophylactic nebulized antibiotics can prevent ICU-acquired pneumonia. A meta-analysis of antibiotics administered *via* the respiratory tract was published in 2018 (*Póvoa et al., 2018*). However, the inclusion of studies utilizing both nebulization and intratracheal instillation may have affected their generalizability. Moreover, several studies focusing specifically on nebulized antibiotics for pneumonia prevention have recently been published. Therefore, this meta-analysis aims to include newly published articles that specifically investigated the preventive efficacy of nebulized antibiotics.

## METHODS

### Eligibility criteria

Randomized clinical trials (RCTs) comparing ICU-acquired pneumonia rates between prophylactic nebulized antibiotics and placebos were included. The study population

consisted of critically ill adult patients (aged ≥18 years) without pneumonia at the time of admission. No language restrictions were applied. The exclusion criteria were as follows: (1) non-human studies, (2) studies primarily focused on children, (3) non-RCT studies, (4) studies that used nebulized antibiotics for the treatment of pneumonia, (5) studies with incomplete data, and (6) non-original studies, such as letters, reviews, or editorials. This systematic review and meta-analysis were registered in the PROSPERO database (Registration No. CRD42024505549).

## Search strategy

Two authors (GM and YXX) independently conducted a comprehensive literature search in the Cochrane Library, PubMed, and Embase databases, covering the period from the inception of the databases until January 23, 2024, the date when data retrieval was completed. The detailed search strategy is available in the Supplemental Material (Appendix 1). Additionally, the reference lists of relevant studies were reviewed to ensure the inclusion of all potential publications.

## Data extraction and quality assessment

A standardized data collection form was developed to extract relevant data. Data extraction was independently performed by GM and YXX, with the process was completed on January 30, 2024. In cases of disagreement, a third investigator, the senior author (ZY), was consulted to resolve the differences. The primary endpoint was the incidence of ICU-acquired pneumonia. The secondary endpoints included mortality (defined as ICU, hospital, or 28-day mortality, with preference given to the longest reported follow-up), length of ICU stay, days of mechanical ventilation, side effects, and systemic use of antibiotics.

The quality of the RCTs was independently assessed by GM and YXX using the Cochrane Collaboration tool. Publication bias was evaluated through visual inspection of funnel plots and assessed statistically using Egger's test.

## Statistical analysis

Statistical analyses were conducted using RevMan 5.3 (Cochrane, London, UK) and STATA 14.0 (StataCorp, College Station, TX, USA) software. Binary variables were reported as odds ratios (OR) with 95% confidence intervals (CI), while continuous variables were expressed as weighted mean differences (WMD) with 95% CI. For studies that only provided the median and quartiles but had a sufficiently large sample size, the mean and variance were estimated (*Luo et al., 2018*; *Wan et al., 2014*). The $I^2$ statistic was used to assess statistical heterogeneity. If $I^2 \leq 50\%$ and $P > 0.1$, statistical homogeneity was assumed, and the fixed-effects model was applied. Conversely, if $I^2 > 50\%$ and $P < 0.1$, statistical heterogeneity was considered present, and the random-effects model was used. Sensitivity analyses were performed to confirm the robustness of the meta-analysis results. Funnel plots were generated to visually inspect for publication bias and Egger's regression

test was conducted to assess the symmetry of the funnel plots. A *P*-value < 0.05 was considered indicative of publication bias.

## Risk of bias and GRADE assessment

The risk of bias was evaluated using the Cochrane risk-of-bias tool for randomized controlled trials (RoB 2.0). The quality of evidence for the outcome measures was assessed using the GRADEpro GDT online tool, in which the initial quality of evidence for RCTs was rated as high. Quality was downgraded based on five factors: risk of bias, inconsistency, indirectness, imprecision, and publication bias. Two independent assessors (GM and YXX) conducted the evaluation, and any disagreements were resolved by a third reviewer (ZY).

# RESULTS

## Search results

The literature search initially identified 2,316 articles. After removing 107 duplicates, 2,191 articles were excluded because they were not relevant to the topic of this meta-analysis. An additional 12 articles were excluded for the following reasons: conference abstracts (*n* = 1), meta-analyses (*n* = 2), nebulization used for treatment rather than prevention (*n* = 2), non-RCT studies (*n* = 1), nebulized antibiotics used for the prevention of fungal infections or Pneumocystis pneumonia (PCP) (*n* = 4), and study protocols (*n* = 2). Ultimately, six RCTs were included in this meta-analysis (*Ehrmann et al., 2023*; *Greenfield et al., 1973*; *Rathgeber et al., 1993*; *Wood et al., 2002*; *Claridge et al., 2007*; *Karvouniaris et al., 2015*) (Fig. 1), involving a total of 1,287 patients, of whom 636 received prophylactic antibiotic therapy.

Regarding publication bias, the *P*-value from Egger's test was not significantly different (*P* = 0.172). However, the funnel plot was asymmetric, suggesting a risk of bias in smaller studies without positive results (Appendix, Figs. S1, S2A, S2B).

## Risk of bias and GRADE assessment

Four study outcome was rated as having an overall low risk of bias (RoB). The other study outcomes were rated as having either some concerns or a high overall RoB (Appendix, Fig. S3). The certainty of evidence for the meta-analytic outcomes ranged from moderate to very low (Fig. S4). The main reasons for downgrading the evidence included the risk of bias, inconsistency, and imprecision. We did not downgrade due to publication bias in accordance with the grading criteria. Indirectness was not downgraded as this review adhered strictly to the eligibility criteria for population, intervention, comparator, and outcomes.

The risk of bias was assessed using the Cochrane Risk of Bias Tool for Randomized Controlled Trials (RoB 2.0). To evaluate the quality of evidence for each outcome measure, we used the GRADEpro GDT online tool. By default, RCTs are initially rated as providing high-quality evidence; however, quality may be downgraded based on five key factors: risk of bias, inconsistency, indirectness, imprecision, and potential publication bias. These factors were carefully considered to assign the final evidence rating for each outcome.
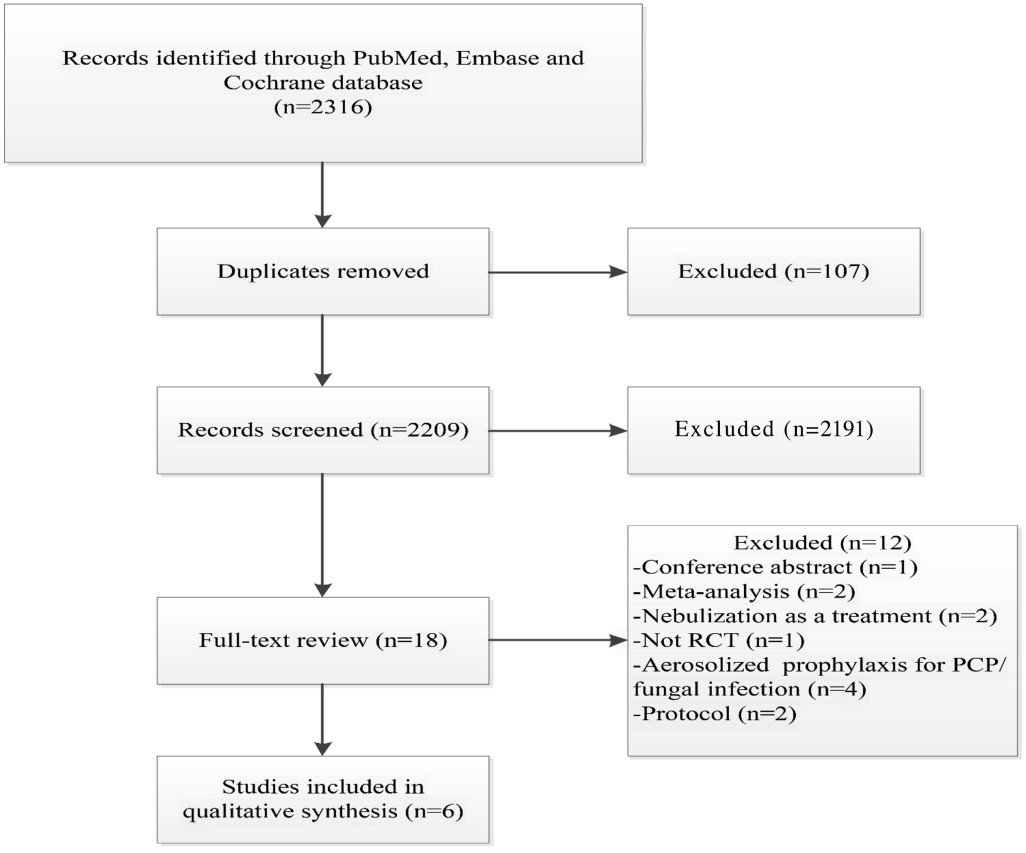

**Figure 1 Flowchart of the selection process for RCTs included in the meta-analysis.**

## Drug administration

Table 1 provides the basic characteristics of the six included studies. The nebulization methods used included the DeVilbiss hand atomizer (*Greenfield et al., 1973*), jet nebulizer (*Rathgeber et al., 1993*; *Wood et al., 2002*; *Claridge et al., 2007*; *Karvouniaris et al., 2015*), and vibrating mesh nebulizer (*Ehrmann et al., 2023*). The antibiotics administered were polymyxin B (*Greenfield et al., 1973*), tobramycin (*Rathgeber et al., 1993*), ceftazidime (*Wood et al., 2002*; *Claridge et al., 2007*), colistimethate sodium (*Karvouniaris et al., 2015*), and amikacin (*Ehrmann et al., 2023*). The control group typically received saline, although two studies did not describe the control group (*Greenfield et al., 1973*; *Rathgeber et al., 1993*). Regarding treatment duration, prophylactic antibiotics were administered until extubation in one study (*Rathgeber et al., 1993*), for 7 days or until extubation in two studies (*Wood et al., 2002*; *Claridge et al., 2007*), for 10 days or until extubation in one study (*Karvouniaris et al., 2015*), and for 3 days in another study (*Ehrmann et al., 2023*), while one study did not specify the treatment duration (*Greenfield et al., 1973*).

Furthermore, five articles reported the use of systemic antibiotics concurrently with nebulized antibiotics. However, due to significant heterogeneity, no pooled analysis was performed. Specifically, two studies described the proportion of systemic antibiotic use. *Greenfield et al. (1973)* demonstrated that during the period of antibiotic nebulization, 88%

**Table 1 Characteristics of studies included in the meta-analysis.**

| Author | Year | Enrolment time | Centers | Study population | Study population | No. of patients | Studied drug/dosage/Drug administration | Mode of administration |
|---|---|---|---|---|---|---|---|---|
| *Greenfield et al. (1973)* | 1973 | December 1970 to June 1972 | Single center | R-SICU | Mechanically ventilated or not | 33 *vs.* 25 | Polymyxin B: A total of 2.5 mg/kg/day was given in six doses, one dose every 4 h (the course of treatment was not mention). | DeVilbiss hand atomizer |
| *Rathgeber et al. (1993)* | 1993 | 4 months (no starting and ending time) | Single center | ICU | Mechanically ventilated | 29 *vs.* 40 | Tobramycin: 80 mg every 6 h until extubation | Jet nebulizer |
| *Wood et al. (2002)* | 2002 | October 1998 to June 2000 | Single center | Trauma ICU | Mechanically ventilated | 20 *vs.* 20 | Ceftazidime: 250 mg every 12 h for 7 days or until extubation or developed VAP. | Jet nebulizer |
| *Claridge et al. (2007)* | 2007 | February 2004 to October 2005 | Single center | Trauma ICU | Mechanically ventilated | 53 *vs.* 52 | Ceftazidime: 250 mg every 12 h for 7 days or until extubation. | Jet nebulizer |
| *Karvouniaris et al. (2015)* | 2015 | November 2011 to July 2013 | Single center | ICU | Mechanically ventilated | 84 *vs.* 84 | Colistimethate sodium: 500000 U, thrice daily for first 10 ICU days or until extubation. | Jet nebulizer |
| *Ehrmann et al. (2023)* | 2023 | July 3 2017 to March 9 2021 | Multicenter | ICU | Mechanically ventilated | 417 *vs.* 430 | Amikacin: 20 mg/kg (ideal body weight) once a day for 3 consecutive days. | Vibrating mesh nebulizer |

| Author | Year | Inclusion criteria | Exclusion criteria | Criteria for pneumonia |
|---|---|---|---|---|
| *Greenfield et al. (1973)* | 1973 | Estimated minimum stay of 72 h in the R-SICU; initiation of aerosol treatment within 24 h of admission to the R-SICU. Neither respiratory failure, tracheostomy, nor intubation with an endotracheal tube was a requirement for inclusion. | Preexisting pneumonia, prior colonization with *P. aeruginosa*, and significant renal failure. | A persistent alveolar infiltrate on at least two chest roentgenograms. Other included evaluation of the gram-stained sputum smear, the sputum culture, temperature of the patient, and the total and differential white blood count. |
| *Rathgeber et al. (1993)* | 1993 | Intubated patients who mechanically ventilated ≥4 days. | Confirmed pneumonia before or after 24 h of intubation; allergy to aminoglycoside therapy. | A new and persistent infiltration on chest radiography; purulent secretions; positive cultures obtained from airway secretion; and at least two of the following criteria: signs of increased respiratory dysfunction and $FiO_2$ increases by more than 0.1; WBC > $10^4/mm^3$; rectal temperature >38 °C |
| *Wood et al. (2002)* | 2002 | Patients ≥16 years old; expected to require at least 7 days of mechanical ventilation; ≥1 risk factors for post-traumatic pneumonia. | Poor prognosis (GCS score of three or expected to survive ≤48 h), allergy to β-lactam, preexistent lung disease requiring long-term inhalation drug therapy, current treatment for a lower respiratory tract infection, long-term therapy with corticosteroids or immunosuppressive drugs, pregnancy, human immunodeficiency virus infection, cancer, or white blood cell count less than 4 × $10^3/mm^3$. | Growth of at least $10^5$ cfu/ml of an organism from BAL in addition to Society of Critical Care Medicine–American College of Chest Physicians criteria for the systemic inflammatory response syndrome. |

| Author | Year | Inclusion criteria | Exclusion criteria | Criteria for pneumonia |
|---|---|---|---|---|
| Claridge et al. (2007) | 2007 | Intubated patients who were believed to require prolonged intubation with a high risk for VAP (The calculated probability of VAP ≥0.25). | Nonsurvivable injuries, ≤18 years old. | Elevated or low white blood cell count, temperature >101.4 °F or <96 °F, infiltrate or change on chest radiograph, or purulent secretions. Quantitative cultures obtained from BAL fluid yielded >$10^5$ CFU of the same organism. |
| Karvouniaris et al. (2015) | 2015 | Age >18 years and mechanical ventilation >48 h. | Grossly purulent sputum or pneumonia on admission, new and persistent infiltrates on chest radiography within 48 h from admission, severe COPD, pregnancy, allergy to colistin, and colonization or infection with a strain resistant to colistin on admission. | VAP diagnosis required (a) the appearance of new or progressive and persistent pulmonary infiltrates on chest radiography and two of the following criteria: (b) abnormal temperature (>38 °C or <36 °C), (c) abnormal WBC count, (d) purulent TBA with a positive culture. TBA culture received within two days before or after clinical diagnosis, that were evaluated by microscopy and considered positive if grew ≥$10^5$ CFU/mL. |
| Ehrmann et al. (2023) | 2023 | Invasive mechanical ventilation ≥72 h. | After 96 h of invasive mechanical ventilation; suspected or confirmed VAP; severe AKI without RRT; CKD; tracheostomy tube; extubation was scheduled within the next 24 h; receiving systemic aminoglycoside therapy. | Positive quantitative bacterial culture in a pulmonary sample and at least two of the following findings: hyperleukocytosis, leukopenia, fever, or purulent secretions with a new infiltrate on a chest radiography. |

| Author | Year | Types of mortality | Types of patients | Time of mechanical ventilation before Enrollment |
|---|---|---|---|---|
| Greenfield et al. (1973) | 1973 | ICU mortality | Respiratory difficulty patients | Not mentioned |
| Rathgeber et al. (1993) | 1993 | Hospital mortality | Study group: Trauma 62%; Surgery 21%; Medical 17% Control group: Trauma 62.5%; Surgery 15%; Medical 22.5% | Mechanical ventilation >4 days |
| Wood et al. (2002) | 2002 | Hospital mortality | Trauma | Not mentioned |
| Claridge et al. (2007) | 2007 | 7-day mortality and hospital mortality | Trauma | Not mentioned |
| Karvouniaris et al. (2015) | 2015 | ICU mortality and hospital mortality | Medical 35.1%; Surgical 26.8%; Neurosugical 38.1% | Mechanical ventilation >48 h |
| Ehrmann et al. (2023) | 2023 | ICU mortality and hospital mortality | Study group: Medical 86%; Scheduled surgery 2%; Unscheduled surgery or trauma 12% Control group: Medical 87%; Scheduled surgery 1%; Unscheduled surgery or trauma 12% | 96 h < mechanical ventilation >72 h |

**Note:**

ICU, Intensive Care Unit; R-SICU, Respiratory-Surgical ICU; VAP, Ventilator-associated pneumonia; WBC, White blood cell; GCS, Glasgow coma scales; BAL, bronchoalveolar lavage; COPD, Chronic obstructive pulmonary disease; TBA, Tracheobroncheal aspirates; AKI, Acute kidney injury; CKD, Chronic kidney disease; RRT, Renal-replacement therapy.

of the cases in the study group and 76% of the cases in the control group received systemic antibiotic therapy (study group: 19/25 *vs.* control group: 29/33). *Rathgeber et al. (1993)* pointed out that there were five cases in the treatment group and two cases in the control group who did not receive systemic antibiotics (study group: 24/29 *vs.* control group: 38/40). *Karvouniaris et al. (2015)* concluded that 124 patients received systemic antibiotics targeting gram-negative bacteria during the 10-day prophylaxis period, with 76.2% (64/84) in the study group and 71.4% (60/84) in the control group, showing no statistically significant difference ($P$ = 0.60). *Wood et al. (2002)* reported the proportion of systemic antibiotic use among patients with ICU-acquired pneumonia (study group: 6/6 *vs.* control group: 11/13) as well as the duration of systemic antibiotic therapy (study group: 15 ± 9 days *vs.* control group: 19 ± 11 days), with both outcomes being statistically insignificant. Finally, *Ehrmann et al. (2023)* reported the number of days when at least one systemic antibiotic was administered and found no significant difference between the study and control groups.

In addition, for patients treated with nebulized antibiotics, no drug concentrations were detected in the blood (*Wood et al., 2002*), and only very low concentrations were detected in a few patients' blood samples (*Greenfield et al., 1973*; *Wood et al., 2002*).

## ICU-acquired pneumonia and multidrug-resistant pneumonia

Six RCT studies ($n$ = 1,287) were included in the analysis. Due to the low heterogeneity ($I^2$ = 14%, $P$ = 0.33), a fixed-effects model was used. The results showed that, compared with the control group, the prophylactic use of nebulized antibiotics significantly reduced the incidence of ICU-acquired pneumonia (OR = 0.57, 95% CI [0.43–0.74], fixed-effects model, six RCTs; Fig. 2), and the result was statistically significant. Sensitivity analysis revealed that the results were robust (see Appendix, Fig. S5A).

Three studies described pneumonia caused by drug-resistant bacteria, but due to differences in reporting formats, no meta-analysis was performed. Specifically, *Greenfield et al. (1973)*, *Rathgeber et al. (1993)*, and *Wood et al. (2002)* described the pathogens in patients with pneumonia, although multidrug resistance was not reported. *Ehrmann et al. (2023)* did not focus on MDR pneumonia but found no differences in routine bacteriological samples in terms of bacteria with acquired resistance to amikacin. Additionally, two studies reported the likelihood of multidrug-resistant (MDR) ventilator-associated pneumonia (VAP). *Claridge et al. (2007)* reported that the percentage of patients with MDR-VAP was 23% in the placebo group and 28% in the ceftazidime group, with no statistically significant difference between the two groups. However, *Karvouniaris et al. (2015)* noted that the incidence of MDR-VAP in the colistin prophylactic nebulization group was significantly lower than that in the control group (study group: 6/84 *vs.* control group: 16/84, $P$ = 0.04). Collectively, there was no clear evidence that prophylactic nebulized antibiotics induced the occurrence of MDR bacterial pneumonia.

**Figure 2 Odds ratios of intensive care unit-acquired pneumonia between patients who received antibiotic prophylaxis and those who received placebo** (*Greenfield et al., 1973*; *Claridge et al., 2007*; *Karvouniaris et al., 2015*; *Rathgeber et al., 1993*; *Ehrmann et al., 2023*; *Wood et al., 2002*).

## Length of ICU stay and mechanical ventilation days

Five articles described the length of ICU stay, although one of them only provided the average and median length of ICU stay (*Greenfield et al., 1973*), and no data conversion was performed. Additionally, six articles reported the number of mechanical ventilation days, with one study reporting only the average and median (*Greenfield et al., 1973*) and another reporting only the average (*Rathgeber et al., 1993*), which could not be included in the analysis. Therefore, four articles were ultimately included in the analysis. Due to the low heterogeneity of both study endpoints ($I^2 = 0\%$, $P = 0.6$ for ICU stay and $I^2 = 0\%$, $P = 0.43$ for mechanical ventilation days), a fixed-effects model was used for the analysis. The results showed that the prophylactic use of antibiotics did not reduce the length of ICU stay (Appendix, Fig. S6) or the number of mechanical ventilation days (Appendix, Fig. S7). Sensitivity analysis confirmed that the results were robust (see Appendix, Fig. S5B and S5C).

## Mortality

Different studies have reported various mortality endpoints, including 7-day mortality (*Claridge et al., 2007*), ICU mortality (*Ehrmann et al., 2023*; *Greenfield et al., 1973*; *Karvouniaris et al., 2015*), and hospital mortality (*Ehrmann et al., 2023*; *Rathgeber et al., 1993*; *Wood et al., 2002*; *Claridge et al., 2007*; *Karvouniaris et al., 2015*). According to pre-established rules, the longest reported follow-up mortality was selected as the endpoint for this study. The findings indicated that prophylactic nebulized antibiotics did not improve patient mortality rates (mortality: OR = 0.86, 95% CI [0.68–1.10], fixed-effects model, six RCTs; Fig. 3). Sensitivity analysis confirmed that the results were robust (see Appendix, Fig. S5D). However, one study reported that prophylactic nebulized antibiotics significantly reduced ICU mortality in patients with ICU-acquired pneumonia (experimental group: 1/14 *vs.* control group: 11/25, $P = 0.028$) (*Karvouniaris et al., 2015*).

**Figure 3** Odds ratios of mortality between patients who received antibiotic prophylaxis and those who received placebo (*Greenfield et al., 1973*; *Claridge et al., 2007*; *Karvouniaris et al., 2015*; *Rathgeber et al., 1993*; *Ehrmann et al., 2023*; *Wood et al., 2002*).

## Side effects

Of the six studies, four reported the side effects of nebulization (*Ehrmann et al., 2023*; *Greenfield et al., 1973*; *Wood et al., 2002*; *Karvouniaris et al., 2015*), with two studies indicating no notable side effects in either the experimental or control groups (*Greenfield et al., 1973*; *Wood et al., 2002*). The side effects in the other two studies were primarily related to respiratory tract disorders (*Ehrmann et al., 2023*; *Karvouniaris et al., 2015*); however, no statistically significant differences were found between the experimental and control groups. These four studies were summarized; however, due to the high heterogeneity ($I^2$ = 62%, $P$ = 0.11) and limited number of studies, subgroup analysis was challenging, leading to the use of a random-effects model. The results showed no significant difference between the nebulized antibiotic and control groups (Appendix, Fig. S8). Sensitivity analysis revealed that the robustness of the study endpoint was relatively weak (see Appendix, Fig. S5E). Upon further examination, it was noted that two of the four studies reported no adverse events, while the largest study (*Ehrmann et al., 2023*) (417 *vs.* 430 participants) also found no significant differences in side effects between the two groups. Therefore, the final results of this analysis were considered relatively reliable.

## DISCUSSION

The incidence of pneumonia in critically ill patients admitted to the ICU is noticeable (*Rosenthal et al., 2024*), and it seriously affects patient prognosis, as well as increases the length of ICU stay and treatment costs (*Warren et al., 2003*; *Rosenthal et al., 2023*). Previous studies have demonstrated that, whether in animal models or clinical studies, the utilization of nebulized antibiotics could promote systemic antibiotic treatment of severe pneumonia, and they exhibited satisfactory therapeutic efficacy in mechanically ventilated patients (*Qin et al., 2021*; *Xu et al., 2018*; *Li Bassi et al., 2019*). As the colonization of pathogens in the upper respiratory tract may constitute the basis for ICU-acquired pneumonia, studies have confirmed that nebulization with antibiotics can reduce pulmonary bacterial colonization and the intensity of inflammatory reactions (*Wood et al.,*

2002; *Ferrari et al., 2009*). The present systematic review and meta-analysis aimed to evaluate the effects of prophylactic nebulized antibiotics on ICU-acquired pneumonia.

ICU patients often experience cough reflex inhibition and reduced tracheal mucosal ciliary movement, which promotes pathogen colonization. Tracheal intubation and positive pressure ventilation further facilitate the movement of colonizing bacteria to the distal airways. Intratracheal administration of antibiotics allows for a high local drug concentration, surpassing the minimum inhibitory concentration (MIC) of the pathogen. The prophylactic use of antibiotics in the bronchial tree reduces airway bacterial colonization without significantly increasing systemic blood drug concentrations, thereby minimizing drug-related organ damage. Hence, this approach may be considered for patients with organ dysfunction. Previous studies have investigated both nebulization and intratracheal instillation as methods of administration (*Póvoa et al., 2018*; *Falagas et al., 2006*). However, simple intratracheal instillation has limitations in drug delivery range compared with nebulization, which forms aerosol inhalation (with droplets of 3.0 to 5.0 μm) capable of reaching the terminal bronchioles and possibly the alveoli (*Wong, Dudney & Dhand, 2019*; *Myrianthefs et al., 2023*). Therefore, the present meta-analysis exclusively included studies related to nebulized antibiotics.

Animal experiments revealed that nebulization of kanamycin before tracheal instillation of *Klebsiella pneumoniae* could prevent bronchopneumonia, while systemic application of antibiotics exhibited no related effect (*Berendt, Magruder & Frola, 1980*). The prophylactic administration of nebulized antibiotics has noticeably attracted clinicians' attention, accompanied by the successful publication of RCTs. Notably, a multicenter, double-blind, randomized, controlled superiority trial was recently published in the New England Journal of Medicine (*Ehrmann et al., 2023*). The findings of this trial indicated that the prophylactic use of nebulized antibiotics could reduce the incidence of VAP. A meta-analysis of six RCTs was undertaken in this study to further investigate this topic, and the results revealed that prophylactic administration of nebulized antibiotics could significantly decrease the incidence of ICU-acquired pneumonia.

Notably, both the prophylactic and therapeutic use of nebulized antibiotics have not yet been included in clinical guidelines (*Dodek et al., 2004*; *Rello et al., 2017*). This exclusion could be attributed to concerns regarding the potential induction of bacterial resistance and the risk of adverse events, particularly respiratory complications. Although these concerns are valid, the evidence primarily stems from non-comparative trials and observational studies, resulting in a low level of evidence and inherent bias. Regarding the potential induction of multidrug-resistant (MDR) bacteria through nebulized antibiotic use, data from randomized controlled trials (RCTs) were insufficient for conducting a meta-analysis on VAP-MDR in this study. However, none of the available literature suggested that nebulized antibiotics increase the incidence of VAP caused by MDR pathogens. Some studies have indicated that prophylactic antibiotic use may reduce the occurrence of drug-resistant bacteria (*Karvouniaris et al., 2015*). This could be due to the short-course administration of prophylactic nebulized antibiotics, which reduces pathogen colonization in the bronchial tree, controls bacterial spread before pneumonia develops, and thereby limits the exposure time to systemic antimicrobial agents (*Ehrmann et al.,*

*2023*). Consequently, this reduction in the exposure time may decrease the likelihood of inducing drug-resistant bacteria. However, this hypothesis remains speculative, as data heterogeneity prevents any conclusive analysis.

Concerns regarding the safety of nebulized antibiotics have long been a focus for clinicians, particularly regarding the potential for airway spasms and increased respiratory resistance. Previous meta-analyses have suggested that nebulized antibiotics, such as amikacin and colistin, could increase the incidence of airway spasms (*Qin et al., 2021*; *Zhang et al., 2023*), particularly when intravenous rather than nebulized formulations are used. However, the present meta-analysis did not find any significant evidence of adverse events. Recent studies have shown that the incidence of serious adverse effects following a 3-day course of nebulized amikacin was less than 2%. Thus, for critically ill patients in the ICU, short-term prophylactic nebulized antibiotics may reduce the incidence of VAP while minimizing airway-related side effects.

As observed in other studies, the use of prophylactic nebulized antibiotics did not result in reduced mortality (*Póvoa et al., 2018*; *Falagas et al., 2006*). This may be partly due to the limited number of articles and cases included in this analysis, reflecting the scope of available RCTs on prophylactic nebulized antibiotics. Additionally, as advancements in medical treatments and supportive care have progressed, the overall incidence of VAP has steadily declined (*Rosenthal et al., 2024*; *Álvarez-Lerma et al., 2018*). In critically ill patients, prognosis is influenced by multiple factors, including comprehensive supportive care and underlying health conditions. As a result, it is difficult to expect significant improvements in clinical outcomes from a single intervention. Nevertheless, one study showed that in patients with VAP, nebulized antibiotics could significantly reduce ICU mortality (*Karvouniaris et al., 2015*). This finding suggests that while prophylactic nebulized antibiotics may not improve overall mortality in the general ICU population, further research is warranted to determine whether they might enhance survival in high-risk groups prone to developing VAP.

In addition, this meta-analysis assessed the length of ICU stay and mechanical ventilation days, but no significant results were obtained. Similar to mortality outcomes, successful removal of tracheal intubation and discharge from the ICU depends on many factors beyond VAP prevention. This is particularly relevant for patients admitted to respiratory surgical ICUs and trauma ICUs who may have pre-existing respiratory conditions or lung injuries. Consequently, it may be unrealistic to expect positive outcomes solely from prophylactic nebulization in these complex patient populations.

This study had several limitations. Firstly, the included studies span a long period, and the nebulization equipment varied among them. Some studies have suggested that different nebulization equipment significantly affect drug deposition rates, which may influence the results to some extent. However, no significant changes were observed over time in either ICU-acquired pneumonia incidence or mortality rates, mitigating the impact of this limitation. Secondly, due to the lack of related research, no subgroup analysis of different nebulized antibiotics was performed. Given the variation in common bacterial strains causing ventilator-associated pneumonia (VAP) across different hospitals, different types of antibiotics may introduce a degree of heterogeneity into the research results.

Third, the study population exhibited a certain level of heterogeneity, encompassing patients from the respiratory, surgical, trauma, and general ICU. Furthermore, although efforts were made to collect data on the duration of mechanical ventilation before patient enrollment, most studies provided only a time range. Specific information, such as the exact duration of mechanical ventilation prior to enrollment or the location of intubation, was unavailable. Additionally, while some studies noted that some patients received systemic antibiotics in conjunction with prophylactic nebulized antibiotics, they only provided the percentage of such patients. Detailed information, such as the duration of systemic antibiotic use, mechanical ventilation time, and mortality rate, was lacking. This made subgroup analyses particularly challenging. Although heterogeneity is common in meta-analyses, the final results should be interpreted with caution. Finally, among the included literature, the study by *Ehrmann et al. (2023)* included the largest number of cases and was weighed much more in the meta-analysis. We have to admit that this may affect the judgment of the final results. However, it can be seen that among the six studies included, there is a tendency to suggest that prophylactic antibiotics can reduce ICU acquired pneumonia to some extent, regardless of whether there is a significant difference. This suggests the reliability of the conclusion to some degree, and of course, more RCT studies in the future may be vital for the final conclusion. Fourth, concerning publication bias, two studies were published some time ago, making it difficult to obtain relevant information on randomization methods, allocation concealment, and blinding. Consequently, the overall risk of bias for these studies was classified as either high or potentially high. The Grading of Recommendations Assessment, Development and Evaluation (GRADE) evidence results showed that out of the five outcome measures, one was rated as moderate, while the remaining four were rated as low or very low. This indicates that the results of this meta-analysis should be interpreted cautiously and underscores the need for improved quality RCTs on prophylactic nebulized antibiotics. Future clinical trials should adhere to the Cochrane risk-of-bias assessment tool and GRADE evidence grading system for standardized reporting.

Finally, although prophylactic nebulized antibiotic therapy did not significantly affect mortality or ICU length of stay, the absence of a formal economic analysis in terms of health economics is noteworthy and should be addressed in future research.

Future research should not only focus on the effects of prophylactic nebulized antibiotics on ICU-acquired VAP, but also investigate whether prophylactic antibiotics could potentially induce bacterial resistance and whether the types of resistant bacteria change after different antibiotics are nebulized. Although this study did not demonstrate a significant effect of prophylactic antibiotics on ICU patient mortality, this may be due to two factors. First, as previously mentioned, differences in the study populations may indicate that prophylactic nebulized antibiotics could provide greater benefits to high-risk ICU-VAP populations. Second, variations in mortality definitions may play a role, as most studies included in this analysis reported ICU or in-hospital mortality, both of which were influenced by a range of comprehensive treatment measures. The relationship between prophylactic nebulized antibiotics and infection-related mortality requires further exploration through large-scale prospective clinical trials.

## CONCLUSIONS

This meta-analysis demonstrated that prophylactic nebulized antibiotic therapy can effectively prevent ICU-acquired pneumonia in critically ill patients without increasing the risk of respiratory infections caused by multidrug-resistant (MDR) pathogens. However, while the reduction in ICU-acquired pneumonia was significant, it did not translate into improvements in mortality or the length of ICU stay.

### Funding

The authors received no funding for this work.

### Competing Interests

The authors declare that they have no competing interests.

### Author Contributions

- Ming Gao performed the experiments, prepared figures and/or tables, and approved the final draft.
- Xiaoxu Yu analyzed the data, prepared figures and/or tables, and approved the final draft.
- Xiaoxuan Liu analyzed the data, prepared figures and/or tables, and approved the final draft.
- Yuan Xu conceived and designed the experiments, authored or reviewed drafts of the article, and approved the final draft.
- Hua Zhou conceived and designed the experiments, authored or reviewed drafts of the article, and approved the final draft.
- Yan Zhu performed the experiments, authored or reviewed drafts of the article, and approved the final draft.

### Data Availability

This is a systematic review and meta-analysis.

### Supplemental Information

Supplemental information for this article can be found online at http://dx.doi.org/10.7717/peerj.18686#supplemental-information.

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
