# Peer review of "Effects of prophylactic nebulized antibiotics on the prevention of ICU-acquired pneumonia: a systematic review and meta-analysis"

_PeerJ, doi:10.7717/peerj.18686_

## Round 0.1 · original submission · Major Revisions

Please address all the review comments.

·

Basic reporting

Clear, unambiguous, professional English was used throughout the manuscript.

Literature references and sufficient field background were provided to support the proposed hypothesis

The article was well-structured with appropriate, figures, and tables.

The relevant results supported the proposed hypotheses.

Experimental design

• Eligibility Criteria:
Inclusion and exclusion criteria were clearly defined, ensuring that the study focuses on relevant, high-quality RCTs.
• Search Strategy:
The description of the search strategy was thorough. However, including the exact search terms and Boolean operators would enhance reproducibility.
• Data Extraction and Quality Assessment:
The data extraction and quality assessment process was robust and involved multiple reviewers and a predefined protocol. Mentioning the resolution of disagreements by a third reviewer has added credibility.
• Statistical Analysis:
The statistical methods were appropriate for a meta-analysis. However, more details on how mean and variance were estimated from medians and quartiles could be beneficial.

Suggestions:
For the Method section:
Include detailed search terms and provide more information on the statistical treatment of median data. For risk of bias assessment, kindly use the Cochrane Collaboration tool for assessing the risk of bias (RoB2) to assess the methodological quality of the RCTs in a tabular form.

Validity of the findings

• Study Selection and Characteristics:
The results section has provided a detailed account of the study selection process and characteristics of the included studies. Including the PRISMA flow diagram has enhanced transparency.
• Primary and Secondary Endpoints:
The findings related to the primary and secondary endpoints were presented with appropriate statistical support.
Publication Bias:
Evaluating publication bias is critical, and the methods used (funnel plots, Egger’s test) are appropriate. Mentioning the lack of significant publication bias has strengthened the study's validity.

Additional comments

Overall Recommendations:
1. Abstract: Briefly mention the types of antibiotics and controls.
2. Introduction: No significant changes are needed; it is comprehensive and well-justified.
3. Methods: Include detailed search terms and provide more information on the statistical treatment of median data. For risk of bias assessment, kindly use the Cochrane Collaboration tool for assessing the risk of bias (RoB2) to assess the methodological quality of the RCTs in a tabular form.
4. Results: Well-written.
5. Discussion: Highlight methodological limitations more explicitly and suggest specific future research directions.
6. Conclusion: well-concluded.

Reviewer 2 ·

Basic reporting

This meta-analysis assessed the efficacy and safety of inhaled antibiotic for preventing ICU-acquired pneumonia and found the potentail benefit of inhaled antibiotics based on the analysis of included six RCTs. Overall the manuscript is well-written.

Experimental design

Please use random effect for all analyses.

Validity of the findings

the number of the study is too limited and most of secondary outcome was based on small number of patients. may add grade assessment.

Additional comments

Nil

·

Basic reporting

The introduction is very clear and understandable. Given that an RCT on prophylactic inhalation of amikacin (PMID: 37888914) has just been published, this systematic review is exceptionally well-timed. The process is appropriate as multiple reviewers utilized three databases. Although the current study did not show an improvement in mortality, it is moving in a favorable direction, and I believe it may be used to significantly improve VAP. However, due to the high heterogeneity among the studies, it would be beneficial to include more narrative descriptions for each RCT, conduct a sensitivity analysis (as mentioned in section 2), and report on Risk of Bias and Grading. While NEJM 2023 showed positive results, other studies also seem to follow a similar trend.

Experimental design

The design as a meta-analysis is good, but it should mention the Risk of Bias (e.g., ROB2) and the Grading of evidence for each study. Is the mortality reported infection-related mortality, all-cause mortality, 28-day mortality, or 90-day mortality? These details should also be added to the table for each individual RCT.

Validity of the findings

The studies have high heterogeneity, as they involved different antibiotics, some included systemic administration, and the time periods ranged from quite old to more recent. To make each study clearer, it is important to describe the baseline characteristics of the target patients. Were they in a surgical ICU, post-cardiac surgery ICU, or a medical ICU? Additionally, how long was the time between hospital admission and intubation? Was it more than 72 hours, or did it occur during hospitalization or in the ER? The longer the duration, the higher the risk of VAP. Given the high heterogeneity, a sensitivity analysis should be conducted by performing a subgroup analysis comparing systemic administration plus nebulization versus nebulization alone.

Additional comments

no

---

## Round 0.2 · Minor Revisions

Please just polish the last few details asked for by reviewer 3

·

Basic reporting

The manuscript was clear, with unambiguous, professional English the language.
The Introduction & background to show context. Literature well referenced & relevant.
The structure conformed to PeerJ standards, and discipline norms, or improved for clarity.
Figures are relevant, high quality, well labelled & described and raw data supplied.

Experimental design

The research question well defined, relevant & meaningful. It is stated how the research fills an identified knowledge gap. Rigorous investigation performed to a high technical & ethical standard. Methods described with sufficient detail & information to replicate.

I raised the following concerns last time;
Q1:Search Strategy: The description of the search strategy was thorough. However, including the exact search terms and Boolean operators would enhance reproducibility.
Q2: Statistical Analysis: The statistical methods were appropriate for a meta-analysis. However, more details on how mean and variance were estimated from medians and quartiles could be beneficial.
Q3 For risk of bias assessment, kindly use the Cochrane Collaboration tool for assessing the risk of bias (RoB2) to assess the methodological quality of the RCTs in a tabular form.

All the questions were addressed.

Validity of the findings

All underlying data have been provided; they were robust, statistically sound, & controlled.

Additional comments

The author has addressed all the previously raised questions. I am happy with the updated manuscript.

Reviewer 2 ·

Basic reporting

The number of included studies was limited and most of findings would be driving by the 2023 study with largeset patient number. In addition, some findings were based on the studies with high heterogeneity. Thus, I cannot have positive recommendation.

Experimental design

Fair

Validity of the findings

Yes

Additional comments

The number of included studies was limited and most of findings would be driving by the 2023 study with largeset patient number. In addition, some findings were based on the studies with high heterogeneity. Thus, I cannot have positive recommendation.

·

Basic reporting

The revisions related to risk of bias, sensitivity analysis, and study limitations have been addressed very well. However, regarding the response:

"Answer: Risk of bias and GRADE assessment to the main text. The results of the GRADE assessment can be found in Appendix 5, Fig. S4."

I could not locate the mentioned "Fig. S4" in the manuscript. Please review the document and ensure that the referenced figure is correctly labeled and included in the supplementary materials.

Experimental design

no comment

Validity of the findings

no comment

Additional comments

no comment

---

## Round 0.3 · accepted · Accept

I think it is a very valuable contribution